# Numerical Investigation of Heat Transfer and Development in Spherical Condensation Droplets

**DOI:** 10.3390/mi15050566

**Published:** 2024-04-26

**Authors:** Jian Dong, Siguang Lu, Bilong Liu, Jie Wu, Mengqi Chen

**Affiliations:** 1Key Laboratory of E&M, Zhejiang University of Technology, Hangzhou 310023, China; 201706041016@zjut.edu.cn (S.L.); 201806040711@zjut.edu.cn (B.L.); 201806040630@zjut.edu.cn (J.W.); 2112102096@zjut.edu.cn (M.C.); 2State Key Lab of Transducer Technology, Shanghai Institute of Microsystem and Information Technology, Chinese Academy of Sciences, Shanghai 200031, China

**Keywords:** condensation droplet, heat transfer, droplet growth, energy functional, dissipative structure, the principle of least action

## Abstract

This study establishes thermodynamic assumptions regarding the growth of condensation droplets and a mathematical formulation of droplet energy functionals. A model of the gas–liquid interface condensation rate based on kinetic theory is derived to clarify the relationship between condensation conditions and intermediate variables. The energy functional of a droplet, derived using the principle of least action, partially elucidates the inherent self-organizing growth laws of condensed droplets, enabling predictive modeling of the droplet’s growth. Considering the effects of the condensation environment and droplet heat transfer mechanisms on droplet growth dynamics, we divide the process into three distinct stages, marked by critical thresholds of 10^5^ nm^3^ and 10^10^ nm^3^. Our model effectively explains why the observed contact angle fails to reach the expected Wenzel contact angle. This research presents a detailed analysis of the factors affecting surface condensation and heat transfer. The predictions of our model have an error rate of less than 3% error compared to baseline experiments. Consequently, these insights can significantly contribute to and improve the design of condensation heat transfer surfaces for the phase-change heat sinks in microprocessor chips.

## 1. Introduction

Due to the excellent heat transfer properties condensation droplets demonstrate on superhydrophobic surfaces, they are used in a wide range of thermal management applications, such as water harvesting, latent heat transfer, refrigeration, electronic thermal management, and power generation [1,2,3,4,5,6,7]. To further enhance the heat transfer performance of these surfaces, it is crucial to study the mechanisms of both heat transfer and droplet growth on subcooling surfaces. In his study, Aref Vandadi optimized two-tier surfaces providing guidance for the structure and texture of superhydrophobic materials; this work significantly contributed to the design of efficient condensation surfaces in heat transfer applications [8]. Moreover, the growth of condensation droplets is dynamic and primarily involves nucleation, direct growth, merging, removal, and re-nucleation processes.

However, the growth mechanism of condensation droplets on superhydrophobic surfaces with microstructures remains a topic of ongoing debate. Using Environmental Scanning Electron Microscopy (ESEM), Eucken et al. [9] observed fluctuations in the contact angle during the growth of partially wetting (PW) droplets and Wenzel droplets. They noted a gradual increase in the apparent contact angle for PW droplets, unlike Wenzel droplets, which underwent several depinning events, leading to an expansion of the wetted area. Notably, the apparent contact angle of the Wenzel droplets ultimately failed to reach the theoretically predicted Wenzel contact angle. Rykaczewski [10] established a constant contact angle (CCA) model and a constant base (CB) growth model of the mechanism of droplet growth and determined the droplet growth rate based on these two models. However, in his research, the condensation surface lacked micro- and nanostructures, indicating that the model may not directly apply to droplet growth on such surfaces.

Dropwise condensation is considered a quasi-static process in thermodynamics, where the system transitions from one steady state or equilibrium state to another. If the droplet growth process can be described by a potential function, then, according to the principle of least action, calculating the minimum of this potential function can clarify the developmental laws governing the growth of condensation droplets.

In this paper, we present a more accurate model for the growth of a single condensation droplet. Our contributions are as follows: (1) Thermodynamic assumptions about the ‘adiabatic evolution’ [11,12] in the growth process of condensation droplets are established, and a mathematical expression of the energy functional of these droplets is developed based on the concept of dissipative structures. The entire system is characterized by three potential fields: the temperature field, the surface tension field, and the gravity field. (2) Based on the principles of kinetic theory, a model of the condensation rate at the gas–liquid interface is derived. Subsequently, the heat transfer rates of droplets can be ascertained from the condensation rate model specific to the gas–liquid interface. (3) After constraining the droplet volume, the intermediate variables within the nonlinear equations are resolved by integrating a heat transfer thermal resistance model for a single condensation droplet into the condensation rate model of the gas–liquid interface. (4) We calculate the energy system function of a condensation droplet for different droplet volumes and obtain the true shape of the condensation droplet based on the principle of least action [13,14]. We predict the growth model at different stages and conduct condensation experiments to verify the model’s predictions. We posit that throughout the entire growth process, the radius of the droplet’s contact circle incrementally expands, ultimately attaining a Wenzel state.

## 2. Theoretical Models

### 2.1. Thermodynamic Assumptions for Condensation Droplet Growth

The heat transfer of dropwise condensation is mainly achieved by condensation droplets. The vapor in the vapor–liquid interface is adsorbed by the condensation droplets, and the latent heat from the phase transformation is transferred to the supercooled substrate through the droplets. The process of droplet condensation growth includes two parts: heat transfer and mass transfer. Notably, the rate of surface mass transfer is much lower than that of droplet internal heat transfer [15,16]. When vapor deposits new water molecules on the droplet’s vapor–liquid interface, simultaneous mass and heat transfer processes occur, with the latter being completed instantaneously. Consequently, the temperature field distribution within the droplet rapidly returns to a steady state, devoid of any internal heat sources. This process is characterized in physics as ‘adiabatic evolution.’ Furthermore, the rate of change in the thickness of the deposited layer at the vapor–liquid interface is exceedingly slow over time, thus qualifying it as a gradual invariant.

According to the theory of the ‘adiabatic phenomenon’, the process of increasing the volume of a condensation droplet proceeds as follows: Initially, both the mass and the latent heat from the phase transition are simultaneously applied to the surface of the droplet. Then, the latent heat from the phase transition is rapidly and steadily transferred. Ultimately, the droplet remains in adiabatic equilibrium until the end of the process. The study of the heat transfer process from a condensing droplet at a designated instantaneous volume follows the assumption that the internal heat transfer within the droplet exhibits steady-state thermal conductivity and that there is no internal heat source.

The dynamic process of condensation droplet growth follows the stationary action principle. To describe non-equilibrium thermodynamic phenomena via the variational method, Onsager [17] proposed a variational principle for steady-state heat transfer based on the concept of entropy production, namely, the stationary action principle. Prigogine [18] used the theory of dissipative structures to portray the thermodynamic behavior of systems far from equilibrium. When the dissipative structure is within a small range near the thermodynamic equilibrium, it is possible to derive its stability and kinetic equations from a potential function.

A single condensation droplet is a dissipative structure present in a system. When the droplet is in adiabatic equilibrium, a fine equilibrium structure exists for the dissipative structure (which occupies most of the process). The fine equilibrium structure has two superimposed states: the Cassie state and the Wenzel state. Its energy can be expressed as a general function. The minimum value of the energy generalization for a defined droplet volume can be obtained according to the direct method of variational differentiation. The shape of the droplet is the “true shape” of the droplet when the energy generalization is minimized. Thus, apparent parameters such as the contact angle and the contact line length of the droplet at a constant volume can be obtained from this “true shape”.

The thermal conduction process of the droplet satisfies the differential heat conduction equation [19]:(1)$$\rho c\frac{\partial T}{\partial t}=\lambda \left(\frac{\partial^{2}T}{\partial x^{2}}+\frac{\partial^{2}T}{\partial y^{2}}+\frac{\partial^{2}T}{\partial z^{2}}\right)+\Phi$$
where *ρ* is the density of the liquid, *c* is the specific heat capacity, *λ* is the thermal conductivity of the liquid, and Φ is the internal heat source.

According to the assumption that the thermal conduction inside the droplet is in a steady state and that there is no internal heat source, the differential equation for thermal conductivity simplifies to the following:(2)$$\lambda \left(\frac{\partial^{2}T}{\partial x^{2}}+\frac{\partial^{2}T}{\partial y^{2}}+\frac{\partial^{2}T}{\partial z^{2}}\right)=0$$

The constant temperature boundary conditions are $T=T_{0}$.

### 2.2. Mathematical Formulation of Energy Functional for Dissipative Structures

When the droplet is in adiabatic equilibrium, the dissipative structure is a fine equilibrium structure (which it occupies during most of the process). There are two superimposed states in the fine equilibrium structure, the Cassie state and the Wenzel state, whose energies can be represented by a functional. For the energy functional of the droplet under a fixed volume, the minimum value can be obtained by the direct variational method, and the shape of the droplet at the minimum energy functional value is the true shape of the droplet. Consequently, apparent characteristics such as the contact line length and contact angle of the droplet can be obtained. By comparing the minimum energies of the Cassie and Wenzel states, the lower energy state can be identified, which elucidates the transition mechanism between the Cassie and Wenzel states.

The internal energy change (or enthalpy change) caused by the redistribution of the temperature field in the heat conduction process in this system is defined as the thermal potential, as shown in Figure 1. When the droplet is in state 1, the external pressure is *P*, the volume is *V*, the internal uniform temperature field is *T*_s_, the enthalpy is *H*_1_, and the internal energy is *U*_1_. Similarly, when the droplet is in state 2, the external pressure is still *P*, the volume is still *V*, the internal temperature field is *T(x*, *y*, *z)*, the enthalpy is *H*_2_, and the internal energy is *U*_2_. Introducing excess temperature $\Theta (x,y,z)=T(x,y,z)-T_{s}$, the energy increment of state 2 is
(3)$$\Delta E=U_{2}-U_{1}=\Delta H=\underset{\Omega }{∭}\rho c\Theta \left(x,y,z\right)dxdydz$$

Taking the internal energy of the droplet in state 1 as the zero point of thermal potential, ΔE can be defined as the thermal potential energy of the droplet in state 2:(4)$$E_{a}=\Delta E=\underset{\Omega }{∭}\rho c\Theta \left(x,y,z\right)dxdydz$$

As shown in Figure 2, the system includes a square substrate with a side length of L_0_, using the substrate’s vapor–solid interface energy as the baseline for surface potential. The potential surface energy of the system consists of the outer contour of the droplet and the solid–liquid contact surface. The surface potential energies of the Wenzel state and the Cassie state can be expressed as follows:(5)$$E_{b-w}=2\pi r^{2}\sigma_{\mathrm{lv}}\left(1-\cos\theta \right)+r_{\mathrm{gh}}(\sigma_{\mathrm{sl}}-\sigma_{\mathrm{sv}})\pi r^{2}\sin^{2}\theta$$
(6)$$E_{b-c}=2\pi r^{2}\sigma_{\mathrm{lv}}\left(1-\cos\theta \right)+(\sigma_{\mathrm{sl}}-\sigma_{\mathrm{sv}})\pi r^{2}\sin^{2}\theta$$
where *r* is the droplet radius, *θ* is the contact angle, *r_gh_* is the surface roughness, and *σ* is the surface tension. The subscripts lv, sl, and sv represent the liquid–vapor, solid–liquid, and solid–vapor interfaces, respectively.

The height of a spherical droplet with a radius of curvature *r* and contact angle *θ* is
(7)h=r(1−cosθ)

Therefore, the center of mass of the spherical droplet is located on its central axis, and the distance from the bottom surface is
(8)$$Z_{c}=\frac{(4r-h)h}{12r-4h}=\frac{r(3+\cos\theta )(1-\cos\theta )}{8+4\cos\theta }$$

The bottom of the microstructure serves as the zero point of the gravitational potential, and the gravitational potential energy of the spherical droplet is
(9)$$E_{c}=\frac{\pi }{12}\rho gr^{4}\frac{\left(2-3\cos\theta +\cos^{3}\theta \right)\left(3-2\cos\theta -\cos^{2}\theta \right)}{2+\cos\theta }$$
where *ρ* is the droplet density and *g* is the gravitational acceleration.

Therefore, the energy functionals of the condensed droplets in the Wenzel state and the Cassie state can be expressed as follows:(10)$$\begin{array}{ll} \Pi_{\mathrm{wenzel}} & =\underset{\Omega }{∭}\rho c\Theta \left(x,y,z\right)dxdydz+2\pi r^{2}\sigma_{\mathrm{lv}}\left(1-\cos\theta \right)+r_{\mathrm{gh}}(\sigma_{\mathrm{sl}}-\sigma_{\mathrm{sv}})\pi r^{2}\sin^{2}\theta \\ & +\frac{\pi }{12}\rho gr^{4}\frac{\left(2-3\cos\theta +\cos^{3}\theta \right)\left(3-2\cos\theta -\cos^{2}\theta \right)}{2+\cos\theta } \end{array}$$
(11)$$\begin{array}{ll} \Pi_{\mathrm{cassie}} & =\underset{\Omega }{∭}\rho c\Theta \left(x,y,z\right)dxdydz+2\pi r^{2}\sigma_{\mathrm{lv}}\left(1-\cos\theta \right)+(\sigma_{\mathrm{sl}}-\sigma_{\mathrm{sv}})\pi r^{2}\sin^{2}\theta \\ & +\frac{\pi }{12}\rho gr^{4}\frac{\left(2-3\cos\theta +\cos^{3}\theta \right)\left(3-2\cos\theta -\cos^{2}\theta \right)}{2+\cos\theta } \end{array}$$

### 2.3. Heat Transfer Model

Figure 3 shows the condensation heat transfer model and simplified thermal resistance network of a single droplet in the Wenzel state and in the Cassie state on a microstructured surface. Because of the droplet’s small size, the heat transfer rate of the gas-liquid phase transition is much smaller than the heat transfer rate of the droplet; therefore, the Marangoni convection inside the droplet is ignored in the model and heat conduction is considered the main mode of heat transfer in the droplet. The micro-pillar height is *δ*, and the surface is modified with a hydrophobic coating of thickness *δhc*. The heat transfer model of a single condensation droplet is simplified to a thermal resistance network, indicating that heat transfer from saturated vapor to the substrate is limited by various thermal resistances, including the vapor–liquid interface’s thermal resistance *Ri*, the thermal resistance of the droplet’s heat conduction *Rd*, the micro-pillar’s thermal resistance *Rp*, and the thermal resistance of the liquid bridges between micro-pillars and the hydrophobic coating *Rhc1*, *Rhc2*.

Therefore, the heat transfer rate through a single Wenzel or Cassie condensation droplet can be derived from G Hu [20]:(12)$$q_{w}=\frac{\Delta T\pi r^{2}}{\left\{\begin{array}{l} \frac{1}{2h_{i}\left(1-\cos\theta \right)}+{\left[\lambda_{\mathrm{hc}}\sin^{2}\theta \left(\frac{\lambda_{p}\varphi }{\delta_{\mathrm{hc}}\lambda_{p}+\delta \lambda_{\mathrm{hc}}}+\frac{\lambda_{l}\left(1-\varphi \right)}{\delta_{\mathrm{hc}}\lambda_{l}+\delta \lambda_{\mathrm{hc}}}\right)\right]}^{-1}\\ +\frac{r}{\lambda_{l}\sin\theta }{\left[\left(1-0.0043\theta )\cot\frac{\theta }{2}-\frac{2}{\theta }\ln(0.0172\theta -0.000074\theta^{2}\right)\right]}^{-1} \end{array}\right\}}$$
(13)$$q_{c}=\frac{\Delta T\pi r^{2}}{\left\{\begin{array}{l} \frac{1}{2h_{i}\left(1-\cos\theta \right)}+\frac{\delta_{\mathrm{hc}}\lambda_{p}+\delta \lambda_{\mathrm{hc}}}{\lambda_{\mathrm{hc}}\lambda_{p}\varphi \sin^{2}\theta }+\frac{r}{\lambda_{l}\sin\theta }\\ •{\left[\left(1-0.0043\theta \right)\cot\frac{\theta }{2}-\frac{2}{\theta }\ln\left(0.0172\theta -0.000074\theta^{2}\right)\right]}^{-1} \end{array}\right\}}$$
where Δ*T* = *Tsat* − *Ts* is the degree of surface subcooling.

On the other hand, we consider condensation heat transfer from the point of view of mass transfer at the vapor–liquid interface. Carey derived an expression for the mass transfer between phases under non-equilibrium conditions based on kinetic theory [21,22]:(14)$$\omega =\left(\frac{2\gamma }{2-\gamma }\right){\left(\frac{\bar{M}}{2\pi \bar{R}}\right)}^{1/2}\left[\frac{P_{v}}{T_{v}^{1/2}}-\frac{P_{i}^{**}}{T_{i}^{1/2}}\right]$$
where *γ* is the condensation coefficient (for pure steam, *γ* = 1; in addition, the usual industrial environment contains a large volume of non-condensable gas, such that *γ* = 0.04), $\bar{M}$ is the molecular weight, $\bar{R}$ is the gas constant, *P*_v_ is the actual steam pressure, *T*_v_ is the steam temperature, *T*_i_ is the vapor–liquid interface temperature, and $P_{i}^{**}$ is the equilibrium pressure corresponding to the vapor–liquid interface temperature *T*_i_.

For spherical condensation droplets, the change in vapor pressure due to the curved vapor–liquid interface can be expressed by the Kelvin–Helmholtz equation:(15)$$\ln\frac{P_{r}^{**}}{P^{*}}=\frac{2\sigma \upsilon_{l}}{r\bar{R}T}$$
where $P_{r}^{**}$ is the equilibrium pressure corresponding to the droplet radius *r*, *P** is the saturation pressure corresponding to the uniform temperature *T* of the system, σ is the surface tension, and *v*_1_ is the specific volume of the droplet. With $P_{r}^{**}$ = $P_{i}^{**}$ at the vapor–liquid interface, the net condensation rate of a droplet of radius *r* is obtained [14,15]:(16)$$\omega =\left(\frac{2\gamma }{2-\gamma }\right){\left(\frac{\bar{M}}{2\pi \bar{R}}\right)}^{1/2}\left[\frac{P_{v}}{T_{v}^{1/2}}-\frac{P_{i}^{*}}{T_{i}^{1/2}}e^{\frac{2\sigma \upsilon_{l}}{r\bar{R}T_{i}}}\right]$$

According to the Clausius–Clapeyron relation,
(17)$$\ln\frac{P_{i}^{*}}{P_{v}^{*}}=-\frac{h_{\mathrm{fg}}}{\bar{R}T_{i}}\left(1-\frac{T_{i}}{T_{v}}\right)$$

the condensation rate of a droplet of radius *r* can be written as
(18)$$\omega =\left(\frac{2\gamma }{2-\gamma }\right){\left(\frac{\bar{M}}{2\pi \bar{R}}\right)}^{1/2}\left[\frac{P_{v}}{T_{v}^{1/2}}-\frac{P_{v}^{*}}{T_{i}^{1/2}}e^{-\frac{h_{\mathrm{fg}}}{\bar{R}T_{i}}\left(1-\frac{T_{i}}{T_{v}}\right)}e^{\frac{2\sigma \upsilon_{l}}{r\bar{R}T_{i}}}\right]$$
where $P_{v}^{*}$ is the vapor saturation pressure, *P*_v_ is the actual vapor pressure, and the relationship between them can be expressed as the relative humidity:(19)$$RH(\%)=P_{v}/P_{v}^{*}\times 100\%$$

From the relationship between the net condensation rate at the vapor–liquid interface, the heat transfer rate can be obtained as follows:(20)$$q=\omega h_{\mathrm{fg}}2\pi r^{2}\left(1-\cos\theta \right)$$
where *h*_fg_ is the latent heat of vaporization.

### 2.4. Numerical Methods

By combining the heat transfer inside the condensation droplet with that of the vapor–liquid interface, the control equation of the heat transfer rate through a single condensation droplet under volume constraints is obtained as follows:(21)$$\left\{\begin{array}{l} V=\frac{\pi }{3}r^{3}\left(2-3\cos\theta +\cos^{3}\theta \right)\\ q=\omega h_{\mathrm{fg}}2\pi r^{2}\left(1-\cos\theta \right)\\ \omega =\left(\frac{2\gamma }{2-\gamma }\right){\left(\frac{\bar{M}}{2\pi \bar{R}}\right)}^{1/2}\left[\frac{P_{v}}{T_{v}^{1/2}}-\frac{P_{v}^{*}}{T_{i}^{1/2}}e^{-\frac{h_{\mathrm{fg}}}{\bar{R}T_{i}}\left(1-\frac{T_{i}}{T_{v}}\right)}e^{\frac{2\sigma \upsilon_{l}}{r\bar{R}T_{i}}}\right]\\ T_{i}=T_{b1}+\frac{q}{\lambda_{l}\pi r\sin\theta }{\left[\left(1-0.0043\theta \right)\cot\frac{\theta }{2}-\frac{2}{\theta }\ln\left(0.0172\theta -0.000074\theta^{2}\right)\right]}^{-1}\\ T_{b1}=T_{s}+q•{\left[\pi r^{2}\sin^{2}\theta \lambda_{\mathrm{hc}}\left(\frac{\varphi \lambda_{p}}{\delta_{\mathrm{hc}}\lambda_{p}+\delta \lambda_{\mathrm{hc}}}+\frac{\left(1-\varphi \right)\lambda_{l}}{\delta_{\mathrm{hc}}\lambda_{l}+\delta \lambda_{\mathrm{hc}}}\right)\right]}^{-1}(\mathrm{Wenzel}\;\mathrm{state})\\ T_{b1}=T_{s}+q•{\left[\pi r^{2}\sin^{2}\theta \varphi \left(\frac{\delta_{\mathrm{hc}}}{\lambda_{\mathrm{hc}}}+\frac{\delta }{\lambda_{p}}\right)\right]}^{-1}(\mathrm{Cassie}\;\mathrm{state}) \end{array}\right.$$
where *r* is the droplet radius, *θ* is the droplet contact angle, $\rho_{1}$ is the droplet density, *h*_fg_ is the latent heat of vaporization, *T*_s_ is the temperature of the silicon substrate’s surface, *T*_v_ is the actual vapor temperature, *T*_i_ is the temperature of the vapor–liquid interface, *T*_b1_ is the temperature at the bottom of the droplet, *T*_b1_ is the vapor saturation pressure, and *P*_v_ is the actual vapor pressure.

The control Equation (21) establishes a relationship between all condensing conditions (*P*_v_, *T*_v_, *T*_s_, *R*H), substrate conditions (*φ*, *δ_hc_*, *δ*), and intermediate variables (*q*, *ω*, *T*_i_, *T*_b1_, *r*, *θ*). Some parameters are nested and unknown, and methods for their numerical calculation are required to solve the nonlinear equation.

For the Wenzel state, the process of calculating unknown parameters in the thermal resistance network is shown in Figure 4. Similarly, the relevant parameters for Cassie condensation droplets can be obtained by employing the calculation formula of the Cassie state T_b1_ in the control equation group.

Under volume constraints, the total energy as a function of the contact angle *θ* is calculated, increasing *θ* by Δ*θ*. The calculation process continues until *θ* reaches 180°, thereby obtaining the space of possible shapes *Ω* [*Ω*1, *Ω*2, …] for a droplet of a given volume. The intermediate variables *T*_b1_ and *T*_i_ are obtained based on the control Equation (21) and are used as the temperature boundary condition of the droplet. Then, the temperature field distribution is solved using ANSYS Fluent 2020 software and used to calculate the thermal potential energy. 

A condensation droplet has different energy functional spaces in the Wenzel and Cassie states, and the corresponding energy functional for the spherical droplet is calculated according to Equations (14) and (15). The transformation law for the wetting state is obtained because the spherical droplet must be in the state of minimum energy. Comparing Πwenzel-min and Πcassie-min at each volume *V* to determine which one is smaller, the true infiltration state of the spherical droplet at volume *V* is obtained.

Based on the principle of least action, the shape when the droplet has the minimum energy is the true shape of the droplet. When the energy functional, as described in Formulas (14) or (15), reaches its minimum, the corresponding droplet shape represents the true form for a given volume *V*. Thus, the changing rules of the contact angle and contact circle radius during the growth of the condensed droplet volume are obtained. Figure 5 illustrates the entire process used to determine the actual infiltration state and the shape of the droplet for a given volume *V*.

## 3. Results and Discussion

### 3.1. Influence of Droplet Volume on the Vapor–Liquid Interface and Bottom Temperature

The influence of Wenzel droplet volume on the vapor–liquid interface is demonstrated in Figure 6, assuming that the vapor temperature *T*_v_ is 283 K, the substrate temperature *T*_s_ on the silicon surface is 273 K, the micro-pillar structure height δ on the silicon surface is 15 μm, the thermal conductivity *λ*_p_ is 0.21 W/(m·K), the solid fraction φ is 0.0641, the hydrophobic coating thickness *δ*_hc_ is 1 nm, and the thermal conductivity *λ*_hc_ is 0.2 W/(m·K).

Under consistent condensation conditions, Figure 6 demonstrates the variations in the vapor–liquid interface and the bottom temperature for Wenzel-state condensation droplets of different volumes. When the contact angles of the condensation droplets are the same, the morphologies of the droplets are similar. However, an increase in droplet volume causes the vapor–liquid interface temperature to approach the vapor’s temperature and the bottom temperature of the droplet to approach the substrate’s temperature. Concurrently, both the temperature of the vapor–liquid interface and the bottom temperature exhibit significant changes in relation to the contact angle. Notably, when the droplet’s contact angle reaches approximately 120°, the disparity between these two temperatures is at its maximum.

### 3.2. Influence of Droplet Volume on Heat Transfer

Under the same condensation conditions and contact angle, Wenzel state condensation droplets of smaller volumes exhibit lower heat transfer rates; however, they display higher average heat flux densities through the droplets, as shown in Figure 7. Since the average heat flux density is related to the contact area of the bottom surface of the droplet, when the contact angle is about 120°, the average heat flux density through the droplet is at its smallest. Typically, the average heat flux density is higher in smaller droplets, which enhances surface heat transfer. This finding is consistent with Tanaka’s [23], who observed that heat transfer in condensation was mainly achieved through smaller droplets.

### 3.3. Self-Organized Growth Mechanism of Condensation Droplets

The mechanism of the self-organized growth of the droplet can be determined by observing the changes in the contact angle and the radius of the contact circle when the volume of the condensed droplet increases. Firstly, the contact angle is changed when the droplet volume is constrained, the relevant parameters in the thermal resistance network (e.g., the vapor–liquid interface temperature and droplet bottom temperature) are calculated, and then the results are combined with a numerical calculation method to determine the energy functional under the condensation droplet dissipation structure. The contact angle corresponding to the minimum value of the system’s energy functional is recorded. Following this, the contact circle radius is determined by calculating the volume for a spherical droplet.

When the supercooling degree ∆*T* is 2 K and 10 K, respectively, the variation in droplet contact angle and a contact circle radius of 10^15^ nm^3^ and below is analyzed. As the droplet’s volume increases, the contact angle and contact circle radius corresponding to the minimum value of the droplet’s energy functional are obtained, as shown in Figure 8.

At the onset of condensation, the contact angle increases rapidly, exceeding the equilibrium contact angle of the microstructured surface (150°). Subsequently, the contact angle oscillates around this equilibrium value. Meanwhile, the radius of the contact circle gradually increases. In the middle stage of condensed droplet growth (10^5^ nm^3^ < V < 10^10^ nm^3^), the droplet contact angle stabilizes at the surface equilibrium contact angle, and the droplet grows in a mode characterized by an increasing contact circle area. Due to vapor pressure and the subcooling effects of the surface, as the condensed droplets enter the middle to late growth stages (V > 10^10^ nm^3^), the droplet contact angle gradually decreases from the equilibrium contact angle and finally remains stable around 120°. At this stage, since the energy functional value for the Wenzel-state droplet is lower than that for the Cassie-state one. It is thus determined that the droplet ultimately adopts the Wenzel state rather than the Cassie state.

Due to the effects of surface supercooling, a droplet’s final stable contact angle cannot be accurately predicted using the Wenzel formula alone. This provides some explanation for the uncertainties observed in the experiments of Enright et al. [24], where the actual contact angle did not match the expected Wenzel contact angle.

Nonetheless, the results at the initial stage of condensation droplet growth are consistent with the experimental observations of Enright et al. [24]. Figure 8 clearly illustrates that the contact angle decreases sharply. This reduction is due to the discontinuous changes in the micro- and nanostructures of the wetted area at the bottom of the droplet. An increase in the pinning area at the droplet’s base results in an immediate reduction in the contact angle. The growth of the droplet does not strictly adhere to a model of constant contact with the substrate or a constant contact angle; instead, it varies dynamically. Moreover, throughout the entire growth process of the condensing droplet, the radius of the contact circle generally increases, with only brief periods where it remains nearly constant. Therefore, the droplet growth model of Rykaczewski [10] exhibits certain limitations.

## 4. Experiments

### 4.1. Fabrication of a Rough Si Surface

In this experiment, microstructures with varying dimensional parameters were engineered onto a Si surface. These microstructures were created on a 4-inch, single-crystal Si substrate using photolithography and Deep Reactive Ion Etching (DRIE) processes. Using a high-depth-of-field microscope (VHX-600, KEYENCE, Osaka, Japan), a top view of the Si surface was inspected, and the micro-pillar diameters and center-to-center distances were accurately measured, as depicted in Figure 9, and remained unchanged, as it serves as a clear transition to the formula that presumably follows:(22)$$f=1+\frac{\pi d^{2}}{4{\left(a+d\right)}^{2}}$$

### 4.2. The Condensation Experiment on the Si Surface with Micro-Pillar Structures

Condensation experiments were performed on the prepared Si surface featuring microstructures. The Si surface was not changed, as it was already clear and properly structured. Figure 10 depicts the software interface used to capture images and compute the contact angle. The droplet contact angle was obtained by calculating the average value of the left and right contact angles.

As shown in Figure 11, pictures 1–4 correspond to four microstructured silicon surfaces with different center-to-center distances. In all instances, the red lines coincided exactly with the yellow lines. Moreover, the blue lines greatly deviated from the yellow lines, meaning our model is more accurate and universal than the ellipse model [25,26,27].

In the experiment measuring the contact angle of droplets on a silicon surface with microstructures, the droplets were prevented from infiltrating the microstructure by the surface tension, and so the silicon surface is in a Cassie state. The intrinsic contact angle of the smooth Si surface was experimentally measured to be about 82°. The theoretical contact angle for the silicon surface with a micro-pillar structure was calculated using the Cassie state formula: $cos\theta_{c}=f\left(cos\theta_{1}+1\right)-1$ [28].

Upon comparing the experimental values listed in Table 1 with theoretical ones, it is deduced that as the center-to-center distance between the micro-pillars increased, the solid phase fraction decreased, and the equilibrium contact angle of the silicon surface increased. The contact angles observed in experiments No. 1 and No. 2 aligned with their theoretical counterparts, while the measured values in No. 3 and No. 4 are smaller than the theoretical values. This discrepancy can be attributed to the increased spacing between micro-columns, which causes the base of the droplet to approach the micro-columns. The gap was wetted, forming a meniscus, so that the actual measured contact angle was smaller than the theoretical value. It can be seen from the measurement of the surface contact angle that microstructures can make the originally hydrophilic surface appear superhydrophobic.

### 4.3. The Condensation Experiment on the Si Surface with Micro-Pillar Structures

Condensation experiments were conducted on the prepared Si surface featuring microstructures. The essential experimental equipment and instruments used are detailed in Table 2.

As shown in Figure 12, in the condensation micro-experiment, the temperature and relative humidity inside the condensation chamber were maintained at 24 ± 0.3 °C and 60 ± 3%, respectively, with a corresponding dew point temperature of 15.8 °C. The temperature of the test surface was maintained at 10 °C. Neglecting the impact of the Si wafer’s thickness on the temperature and assuming an even distribution of surface temperature, the surface was approximately 5.8 K. Under consistent condensation conditions, the dynamic process of liquid droplet condensation on silicon surfaces with four different sets of microstructures was observed and documented.

From Figure 13, it is evident that the condensation droplets on all four silicon surfaces ultimately ended up in the Wenzel state. Notably, Surface 1, which has the smallest micro-pillar spacing, tended to form Wenzel-state droplets most readily. Given the higher heat flux density associated with Wenzel-state droplets, which is conducive to condensation heat transfer, it can be inferred that surfaces with smaller micro-pillar spacing exhibit superior heat transfer performance. However, due to the limited observational capabilities of the experimental apparatus, this study was restricted to capturing only the middle-to-late stages of droplet growth. As the volume of the droplet increases, the contact radius progressively expands, resulting in the manifestation of the Wenzel state. This outcome corroborates our conclusions and validates the reliability of the theoretical model.

## 5. Conclusions

In summary, utilizing kinetic theory, we have developed a condensation rate model for the gas–liquid interface of condensation droplets, which facilitated the calculation of the heat transfer rate of droplets of varying volumes. By considering the constraint of droplet volume, we combined the heat resistance model for individual condensation droplets with the condensation rate model for the gas–liquid interface. Through numerical calculations, we were able to solve for the intermediate variables within this complex, nonlinear system.

We classified the growth of condensation droplets into three distinct stages, with volumes of 10^5^ nm^3^ and 10^10^ nm^3^, enabling a more precise prediction of the droplet growth model. First, in the early growth stage, the contact angle increases rapidly and can surpass the surface equilibrium contact angle. Second, during the middle growth stage, the droplet contact angle remains constant, exhibiting growth patterns which align with the increase in the contact circle area. Third, in the late growth stage, the droplet contact angle gradually decreases from the equilibrium contact angle, ultimately stabilizing around 120°. Throughout the entire growth process, the contact circle radius of the droplet gradually increases, ultimately transitioning into a Wenzel state. This aligns with the experimental observations made by Enright et al. [24]. Additionally, our experiments confirmed the accuracy of the proposed growth model. This study lays a solid theoretical foundation and provides quantitative methods for the design and production of surfaces for use in applications such as biochips and microfluidic chips.

## Figures and Tables

**Figure 1 micromachines-15-00566-f001:**
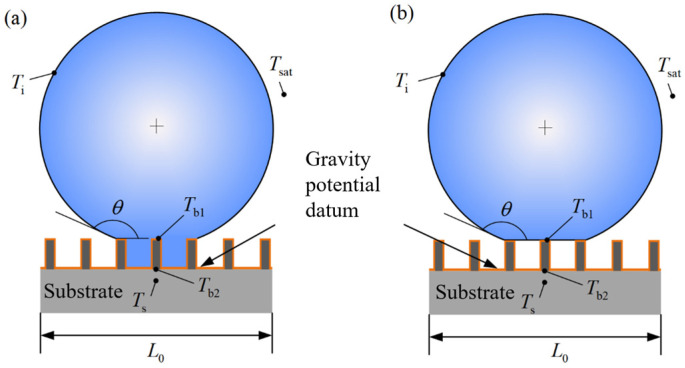
Energy functional system composed of a single spherical condensate droplet and substrate. (**a**) Wenzel state, (**b**) Cassie state.

**Figure 2 micromachines-15-00566-f002:**
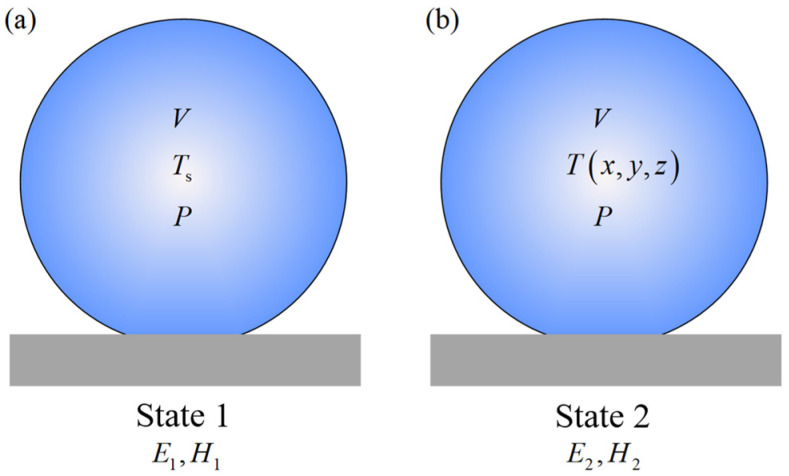
Schematic of the definition of thermal potential. (**a**) State 1, (**b**) State 2.

**Figure 3 micromachines-15-00566-f003:**
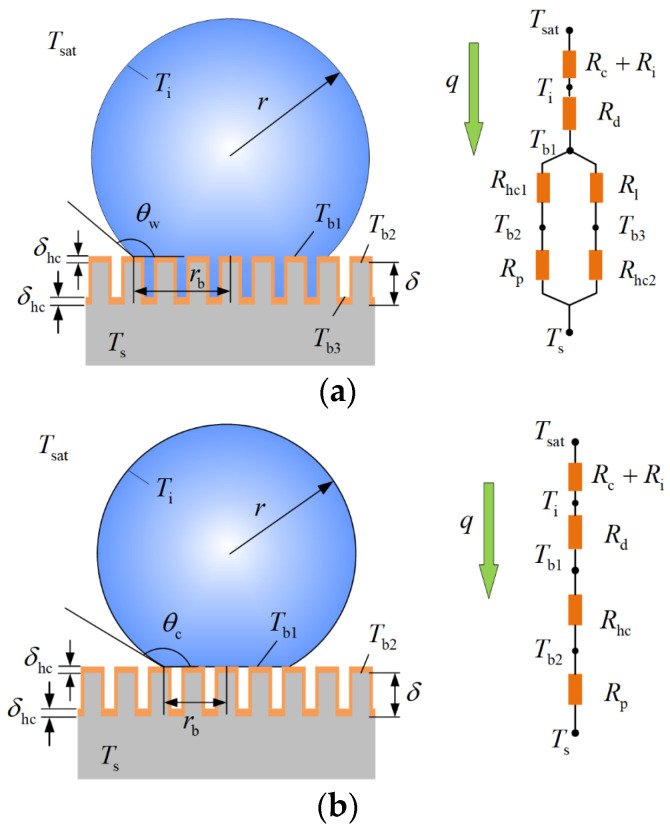
Schematic of the condensation heat transfer model and thermal resistance network of a single droplet in the Wenzel and Cassie states. (**a**) Wenzel state, (**b**) Cassie state.

**Figure 4 micromachines-15-00566-f004:**
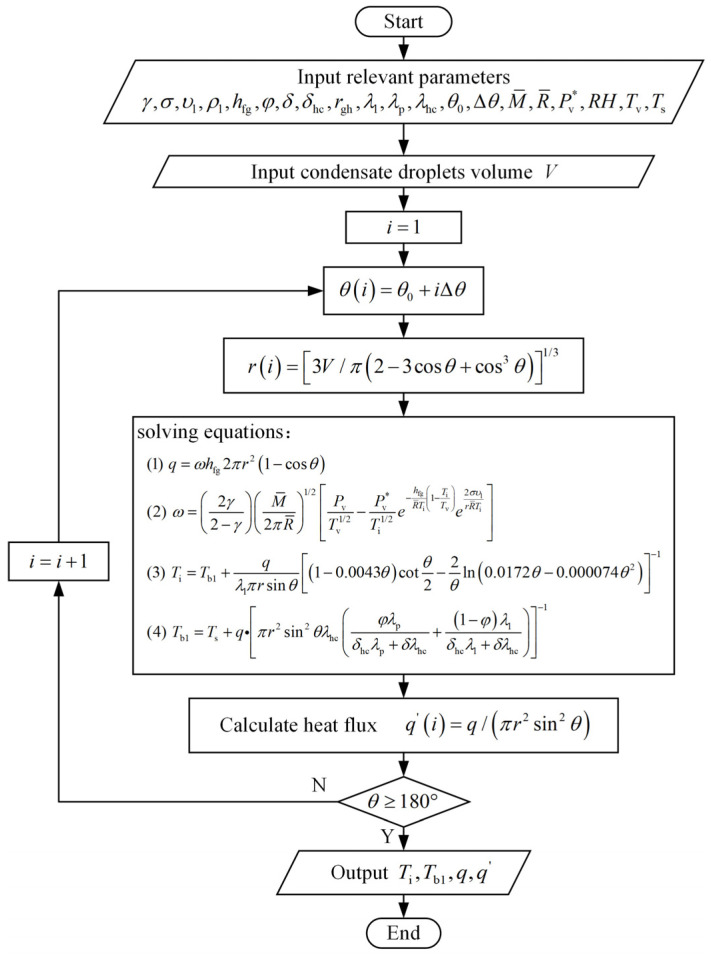
Calculation flow chart of Wenzel state’s intermediate variables.

**Figure 5 micromachines-15-00566-f005:**
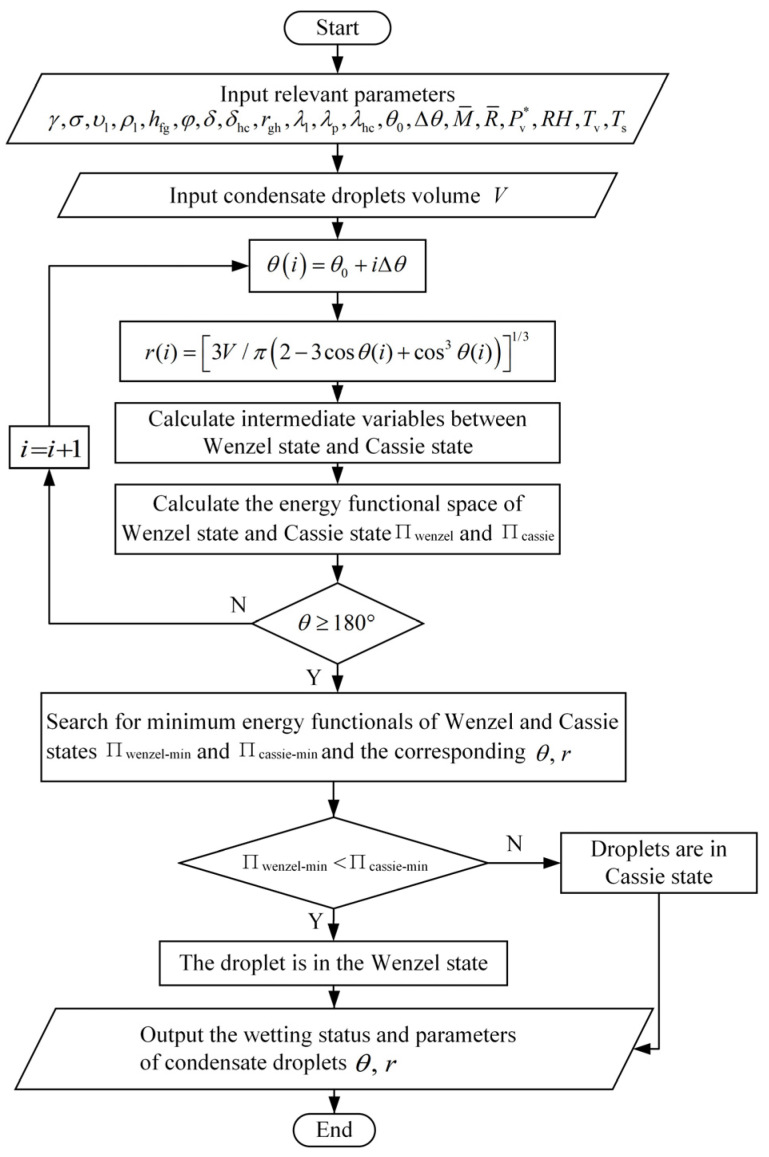
Flow chart for determining wetting state and droplet parameters at volume V.

**Figure 6 micromachines-15-00566-f006:**
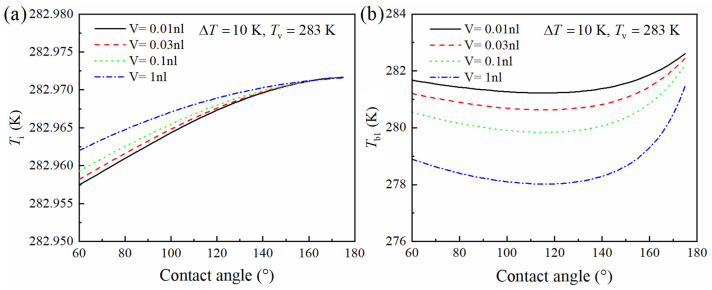
Influence of Wenzel droplet volume on the vapor–liquid interface and bottom temperature. (**a**) Vapor–liquid interface temperature. (**b**) Droplet bottom temperature.

**Figure 7 micromachines-15-00566-f007:**
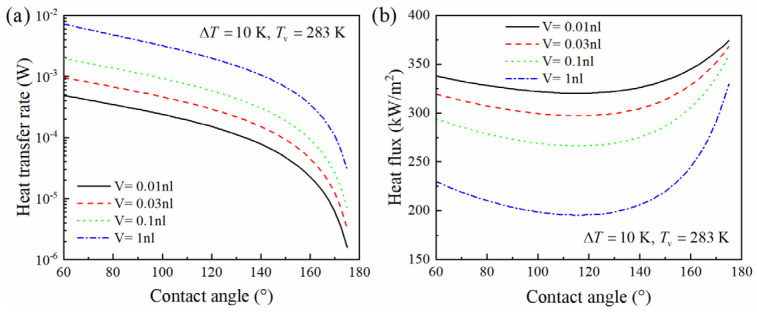
Effect of Wenzel droplet volume on heat transfer. (**a**) Heat transfer rate. (**b**) Heat flux.

**Figure 8 micromachines-15-00566-f008:**
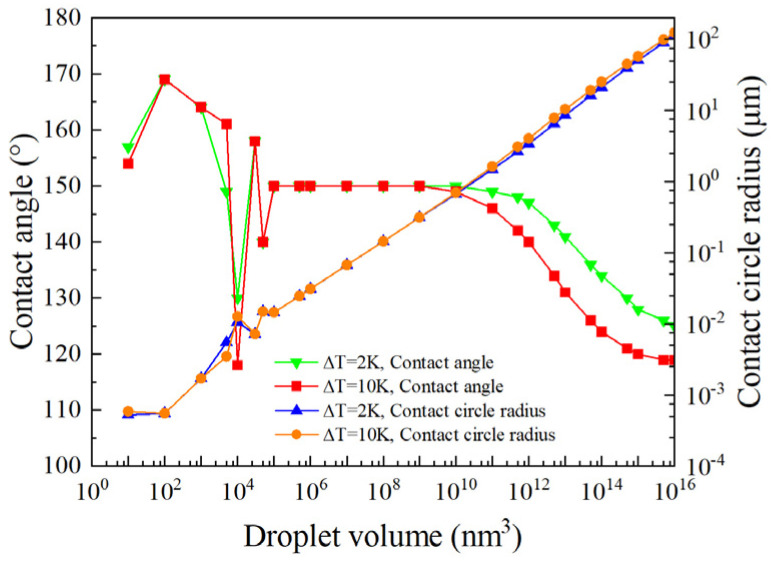
Variation in contact angle and contact circle radius during growth of condensation droplets.

**Figure 9 micromachines-15-00566-f009:**
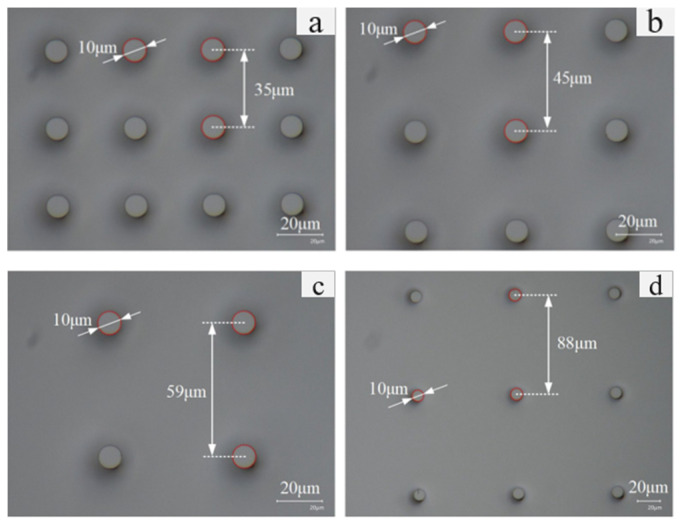
Super-depth-of-field microscope top view of the micro-pillar structure on Si surface. (**a**) No.1, (**b**) No.2, (**c**) No.3, (**d**) No.4.

**Figure 10 micromachines-15-00566-f010:**
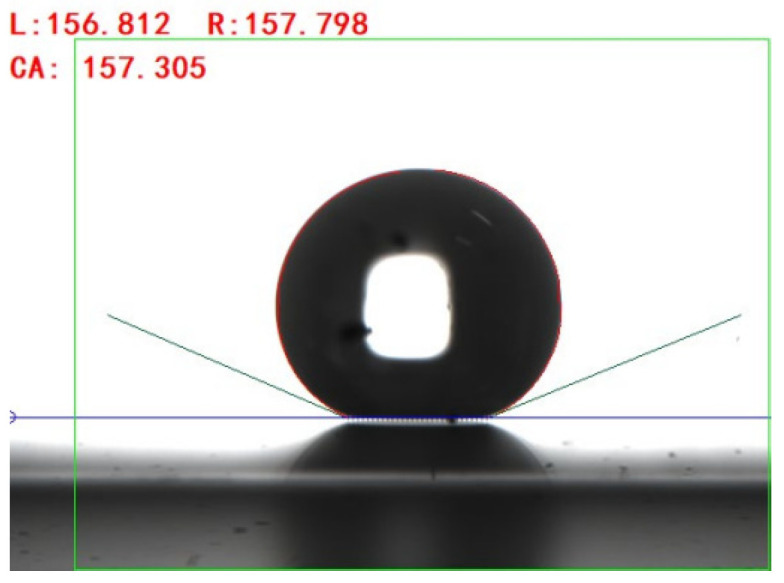
Droplet contact angle measurement interface.

**Figure 11 micromachines-15-00566-f011:**
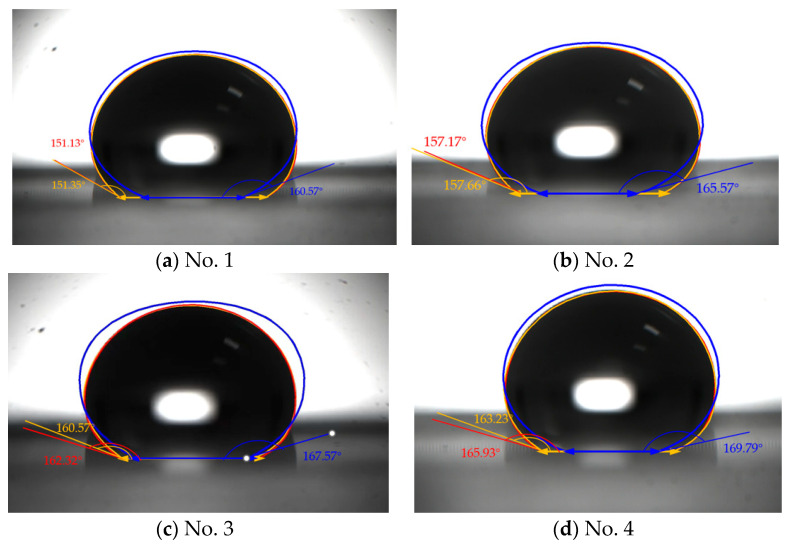
Visual comparison of results from our model, the experiments, and the ellipse model. Profiles from our model are shown with yellow lines, those of the experiments with red lines, and those of the ellipse model with blue lines. The ellipse model is taken from Lubarda’s paper [27].

**Figure 12 micromachines-15-00566-f012:**
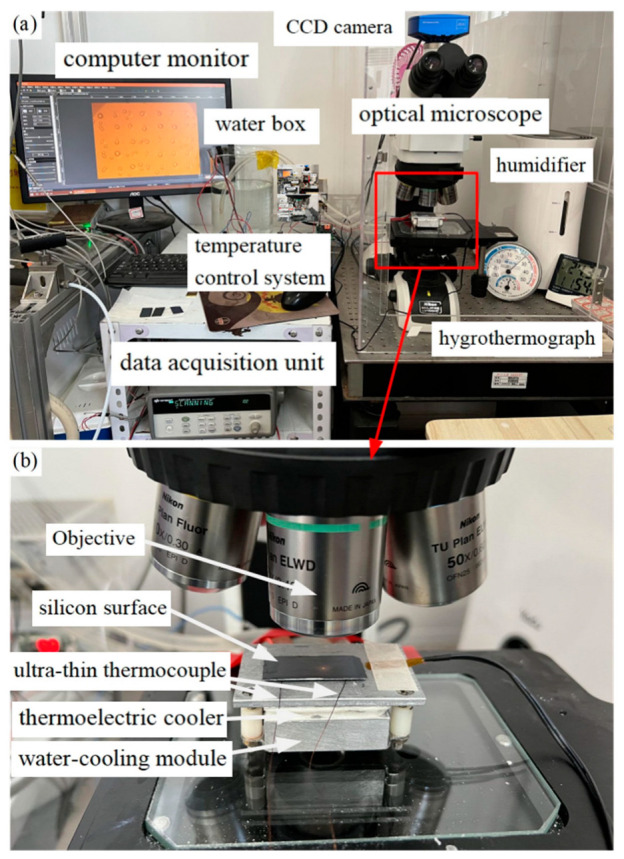
(**a**) Condensation experiment platform. (**b**) shows the magnified observation system.

**Figure 13 micromachines-15-00566-f013:**
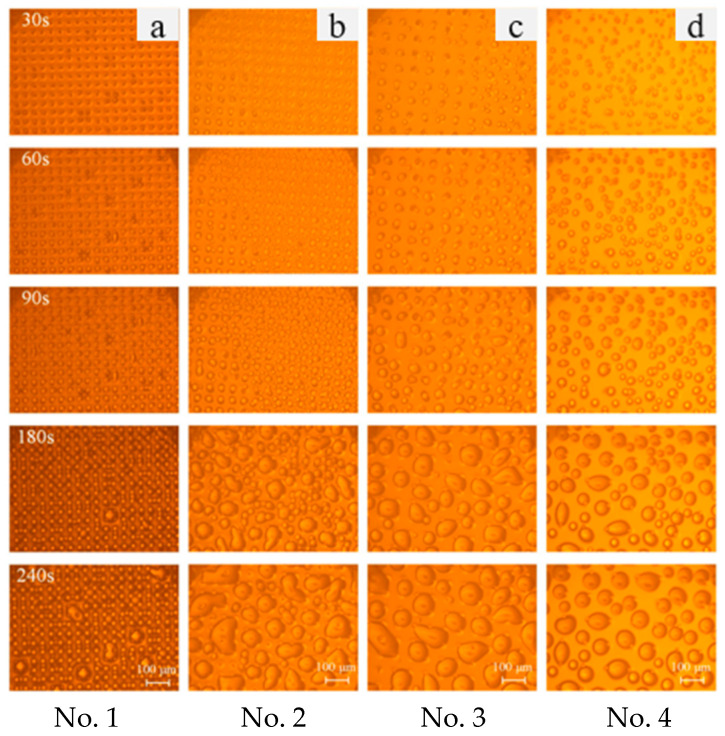
Growth process of condensation droplets on the microstructured silicon surfaces.

**Table 1 micromachines-15-00566-t001:** Static contact angle of micro-pillar-structured silicon surfaces.

Group	*d*/μm	*a*/μm	*h*/μm	*f/φ*	Contact Angle/°	
					Theoretical Value	Experimental Value
1	10	35	15	0.0641	151.13	151.35
2	10	45	15	0.0388	157.66	157.17
3	10	60	15	0.0218	162.32	160.57
4	10	90	15	0.0097	165.93	163.23

**Table 2 micromachines-15-00566-t002:** Equipment details.

Equipment Name	Model	Distributor
Optical microscope	ECLIPSE LV1OOND	Nikon, Shanghai, China
Objective	TU-Plan Flour	Nikon, Shanghai, China
CCD camera	PSC603	Oplenic, Beijing, China
Humidifier	3G40A	Midea, Foshan, China
Hygrothermograph	HTC-1	Purich, Tianjin, China
Peltier semiconductor refrigeration chip	XH-C1201	Xinhe Electronic Technology, Guangzhou, China
Ultrathin thermocouple	T-Type	Benop, Shenzhen, China
Data acquisition unit	34970A	Agilent, Suzhou, China

## Data Availability

The original contributions presented in the study are included in the article, further inquiries can be directed to the corresponding author.

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
