# Peer review of "Numerical Investigation of Heat Transfer and Development in Spherical Condensation Droplets"

_micromachines, 2024, doi:10.3390/mi15050566_

Round 1
Reviewer 1 Report
Comments and Suggestions for Authors
1) The writing needs to be polished. Such as, there are many unclear in the text. Especially the figure numbers are often incorrect.
2) Some equations need to supplement their sources.
3) Additional code validation is required, and the subsequent experimental results need to be compared with that of simulations to demonstrate the accuracy of the model in this manuscript.
4) To add innovation to the manuscript. Because the droplet energy functions method is a common method, the results of this paper need to be proven to be more accurate than those of other literatures.

The writing needs to be polished.
Author Response
|
1) The writing needs to be polished. Such as, there are many unclear in the text. Especially the figure numbers are often incorrect.
|
|
Response 1: Thank you so much for pointing this out. We agree with this comment. Therefore, we polished the text again and corrected expression and numerical errors in lines 14, 36, 233, 65, 305, 409. |
|
2) Some equations need to supplement their sources.
|
|
Response 2: Thank you for your valuable feedback and suggestions. We added the derivation of equation (7) between lines 157-161 and the source of equation (10) in line 186. The specific corrections are as follows: We derive the height h and the distance Zc from the center of mass to the bottom through equations (7) and (8).
Response 3: Thank you for your valuable feedback and suggestions. We have added a comparison chart on page 12 to verify the accuracy of the model with experimental data.
Response 3: Thank you very much for your constructive comments and helping us improve the quality of the manuscript. We add “Error less than 3%” in line 19 to show the maximum error of our model to the experiment while comparing the model of Lubarada et al. on page 12.
|
|
4. Respond to comments about the quality of the English language |
|
point 1:
|
|
Response 1: Thank you for your valuable feedback. We reviewed the manuscript carefully and made necessary language revisions to ensure clarity and flow. Professional English editing services were also hired to ensure the quality of the language. We believe these improvements have addressed your concerns and enhanced the readability of the manuscript. |
|
5. Other instructions |
|
Thank you for your continued support and guidance throughout the review process. |
Reviewer 2 Report
Comments and Suggestions for Authors
In this work, the energy functional of the droplet is derived using the least action principle, which in turn partially elucidates the inherent self-organizing growth laws of condensed droplets, thereby facilitating the establishment of a predictive growth model. I have some concerns on this work:
1. A lot of work have been done in this field and are not properly cited:
Appl. Phys. Lett. 101, 131909 (2012); Nanoscale Advances 1 (3), 1136-1147, 2019
2. The claimed critical thresholds of 10-10 nl and 10-5 nl are order of magnitudes lower than nucleation embryos, which may be not scientific sound.
3. Writing needs significant improvement such as subooling surface.
4. Conclusion should be Conclusions
5. The assumption "Droplet boundaries can be considered adiabatic." is doubtable for an condensation process.
6. Eq (1) is valid for a thermal conduction process, which may limit the applicability of this analysis.
Before the above comments are addressed, I do not recommend its publication in Micromachines.
Comments on the Quality of English LanguageNeed improvements
Author Response
|
3. Point-by-point response to Comments and Suggestions for Authors |
|
|
|
Comments 1: A lot of work have been done in this field and are not properly cited:
|
||
|
Response 1: Thanks for your kind reminding and useful advice. We have thoroughly reviewed the recommended literature and agree that it significantly contributes to the context and depth of our study. Therefore, we have added” Vandadi, Aref … on superhydrophobic surfaces." in lines 31-33.
|
||
|
Comments 2: The claimed critical thresholds of 10-10 nl and 10-5 nl are order of magnitudes lower than nucleation embryos, which may be not scientific sound.
|
||
|
Response 2: Thank you very much for pointing this out. We acknowledge that our initial expression was not sufficiently clear. After consulting additional literature, we understand that the nucleation radius of droplets can range from 1 to 10 nm. The corresponding volume should be between 10-103 nm3. 1 m3 = 1012nl = 10 27 nm3, 10-10 nl = 105 nm3. We believe these results align with scientific standards. Your guidance has been invaluable in identifying areas where our presentation was unclear. Following your suggestion, we have replaced ' nl' with ' nm3' as the unit, thereby rendering our results clearer and more intuitive. We have made corrections in lines 7, 305, 316, 320, 436 and Figure 8.
References: 1. ENRIGHT R, MILJKOVIC N, DOU N, et al. Condensation on superhydrophobic copper oxide nanostructures[J]. Journal of Heat Transfer-Transactions of the ASME, 2013, 135(9): 091304. 2. Tang G, Niu D, Guo L, et al. Failure and recovery of droplet nucleation and growth on damaged nanostructures: A molecular dynamics study[J]. Langmuir, 2020, 36(45): 13716-13724.
Comments 3: Writing needs significant improvement such as subooling surface.
Response 3: Thank you very much for pointing this out. We agree with this comment. Therefore, we have polished the writing again and corrected spelling errors, such as: subooling was changed to subcooling in line 29.
Comments 4: Conclusion should be Conclusions.
Response 4: Thank you for your careful review and attention to detail. We have corrected the heading from 'Conclusion' to 'Conclusions' as suggested.
Comments 5: The assumption "Droplet boundaries can be considered adiabatic." is doubtable for an condensation process.
Response 5: Your constructive comments are highly appreciated and have significantly contributed to the improvement of our manuscript. To clarify and avoid any misunderstanding, we have revised the assumption regarding the adiabatic condition in line 88. The process of droplet condensation is not adiabatic. According to Haken's Slavery Principle in complex systems, the dynamics of a system can typically be categorized into fast and slow variables. In the context of droplet condensation, the rate of heat transfer significantly surpasses that of mass transfer. Therefore, heat transfer can be considered a fast variable because it rapidly reaches a quasi-steady state within a relatively short timeframe. Consequently, the boundary of the droplet can be regarded as experiencing steady-state heat conduction.
Comments 6: Eq (1) is valid for a thermal conduction process, which may limit the applicability of this analysis.
Response 6: Thanks for your kind reminding and useful advice. We recognize the concern that using a steady-state model might introduce limitations. However, we believe that the application of Haken's Slaving Principle and the concept of gradual invariants provides a robust framework that supports our approach. In our study, the rate of heat transfer (a 'fast' variable) significantly exceeds that of mass transfer (a 'slow' variable), allowing the heat transfer process to rapidly reach a quasi-steady state. This disparity in rates justifies approximating the heat conduction at the droplet boundary as a steady-state process.
Furthermore, the concept of adiabatic invariants in the context of Haken's principle supports the notion that certain properties of the system remain constant or change slowly over the time scale of interest, further reinforcing the validity of applying a steady-state approximation in our analysis. By focusing on the dominant fast variable, our approach aligns with the principle that the slow variables are 'enslaved' by the fast variables, reaching a condition where the steady-state approximation provides a reasonable and scientifically justifiable simplification.
Therefore, while we acknowledge the inherent limitations of employing a steady-state heat conduction equation, we argue that within the specific context and dynamics of our study, this approach is supported by well-established theoretical principles. This allows us to capture the essential physics of the process while simplifying the analysis in a manner that is both practical and scientifically sound.
|
||
|
4. Response to Comments on the Quality of English Language |
||
|
Point 1:
|
||
|
Response 1: Thank you for your valuable feedback. We have carefully reviewed the manuscript and made necessary language revisions for clarity and fluency. A professional English editing service was also employed to ensure the quality of the language. We believe these improvements have addressed your concerns and enhanced the readability of the manuscript.
|
||
|
5. Additional clarifications |
||
|
Thank you for your continued support and guidance throughout the review process. |
||
Round 2
Reviewer 2 Report
Comments and Suggestions for Authors
There are two papers in reference 8:
8. Vandadi A, Zhao L, Cheng J. Resistant energy analysis of self-pulling process during dropwise condensation on superhydro-480 phobic surfaces[J]. Nanoscale Advances, 2019, 1(3): 1136-1147.Lubarda, V.A.; Talke, K.A. Analysis of the Equilibrium Droplet 481 Shape Based on an Ellipsoidal Droplet Model. Langmuir 2011, 27, 10705–10713.
Comments on the Quality of English LanguageImproved in the revision
Author Response
Thank you very much for pointing this out. We agree with this comment. We have corrected reference 8 and removed the one of Lubardia and Talke.